# Exploring Smartphone-enabled Text Selection in AR-HMD

Rajkumar Darbar*
Inria Bordeaux, France

Arnaud Prouzeau†
Inria Bordeaux, France

Joan Odicio-Vilchez‡
Inria Bordeaux, France

Thibault Lainé§
Asobo Studio, Bordeaux, France

Martin Hachet¶
Inria Bordeaux, France

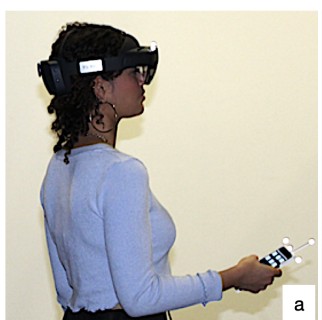
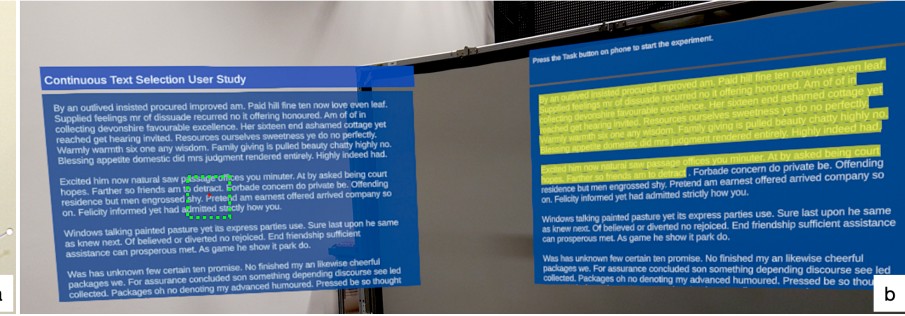

Figure 1: (a) The overall experimental setup consisted of a HoloLens, a smartphone, and an optitrack system. (b) In the HoloLens view, a user sees two text windows. The right one is the 'instruction panel' where the subject sees the text to select. The left is the 'action panel' where the subject performs the actual selection. The cursor is shown inside a green dotted box (for illustration purpose only) on the action panel. For each text selection task, the cursor position always starts from the center of the window.

## ABSTRACT

Text editing is important and at the core of most complex tasks, like writing an email or browsing the web. Efficient and sophisticated techniques exist on desktops and touch devices, but are still under-explored for Augmented Reality Head Mounted Display (AR-HMD). Performing text selection, a necessary step before text editing, in AR display commonly uses techniques such as hand-tracking, voice commands, eye/head-gaze, which are cumbersome and lack precision. In this paper, we explore the use of a smartphone as an input device to support text selection in AR-HMD because of its availability, familiarity, and social acceptability. We propose four eyes-free text selection techniques, all using a smartphone — continuous touch, discrete touch, spatial movement, and raycasting. We compare them in a user study where users have to select text at various granularity levels. Our results suggest that continuous touch, in which we used the smartphone as a trackpad, outperforms the other three techniques in terms of task completion time, accuracy, and user preference.

**Index Terms:** Human-centered computing—Human computer interaction (HCI)—Interaction paradigms—Mixed / augmented reality; Human-centered computing—Human computer interaction (HCI)—Interaction paradigms—Graphical user interfaces;

## 1 INTRODUCTION

Text input and text editing represent a significant portion of our everyday digital tasks. We need it when we browse the web, write emails, or just when we type a password. Because of this ubiquity, it has been the focus of research on most of the platforms we are using daily like desktops, tablets, and mobile phones. The recent

---
*e-mail: rajkumar.darbar@inria.fr
†e-mail: arnaud.prouzeau@inria.fr
‡e-mail: joan.odicio-vilchez@inria.fr
§e-mail: tlaine@asobostudio.com
¶e-mail: martin.hachet@inria.fr

focus of the industry on Augmented Reality Head-Mounted Display (AR-HMD), with the development of devices like the Microsoft HoloLens[1] and Magic Leap[2], made them more and more accessible to us, and their usage is envisioned in our future everyday life. The lack of a physical keyboard and mouse (i.e., the absence of interactive surfaces) with such devices makes text input difficult and an important challenge in AR research. While text input for AR-HMD has been already well-studied [18, 38, 46, 56], limited research focused on editing text that has already been typed by a user. Normally, text editing is a complex task and the first step is to select the text to edit it. This paper will only focus on this text selection part. Such tasks have already been studied on desktop [11] with various modalities (like gaze+gesture [15], gaze with keyboard [51]) as well as touch interfaces [22]. On the other hand, no formal experiments were conducted in AR-HMD contexts.

Generally, text selection in AR-HMD can be performed using various input modalities, including notably hand-tracking, eye/head-gaze, voice commands [21], and handheld controller [34]. However, these techniques have their limitations. For instance, hand-tracking suffers from achieving character level precision [40], lacks haptic feedback [14], and provokes arm fatigue [31] during prolonged interaction. Eye-gaze and head-gaze suffer from the 'Midas Touch' problem which causes unintended activation of commands in the absence of a proper selection mechanism [29, 32, 54, 58]. Moreover, frequent head movements in head-gaze interaction increase motion sickness [57]. Voice interaction might not be socially acceptable in public places [26], and it may disturb the communication flow when several users are collaborating. In the case of a dedicated handheld controller, users need to always carry an extra specific hardware.

Recently, researchers have been exploring to use of a smartphone as an input for the AR-HMD because of its availability (it can even be the processing unit of the HMD [45]), familiarity, social accept-ability, and tangibility [9, 23, 60]. Undoubtedly, there is a huge potential for designing novel cross-device applications with a combination of an AR display and a smartphone. In the past, smartphones

---
[1]https://www.microsoft.com/en-us/hololens
[2]https://www.magicleap.com/en-us

have been used for interacting with different applications running on AR-HMDs such as manipulating 3D objects [41], windows management [47], selecting graphical menus [36] and so on. However, we are unaware of any research that has investigated text selection in an AR display using a commercially available smartphone. In this work, we explored different approaches to select text when using a smartphone as an input controller. We proposed four eyes-free text selection techniques for AR display. These techniques, described in Section 3.1, differ with regard to the mapping of smartphone-based inputs - touch or spatial. We then conducted a user study to compare these four techniques in terms of text selection task performance.

The main contributions of this paper are - (1) design and development of a set of smartphone-enabled text selection techniques for AR-HMD; (2) insights from a 20 person comparative study of these techniques in text selection tasks.

## 2 RELATED WORK

In this section, we review previous work on text selection and editing in AR, and on a smartphone. We also review research that combines handheld device with HMD and large wall displays.

### 2.1 Text Selection and Editing in AR

Limited research focused on text editing in AR. Ghosh et al. presented EYEditor to facilitate on-the-go text-editing on a smart-glass with a combination of voice and a handheld controller [21]. They used voice to modify the text content, while manual input is used for text navigation and selection. The use of a handheld device is inspiring for our work, however, voice interaction might not be suitable in public places. Lee et al. [37] proposed two force-assisted text acquisition techniques where the user exerts a force on a thumb-sized circular button located on an iPhone 7 and selects text which is shown on a laptop emulating the Microsoft Hololens display. They envision that this miniature force-sensitive area (12 mm × 13 mm) can be fitted into a smart-ring. Although their result is promising, a specific force-sensitive device is required.

In this paper, we follow the direction of the two papers previously presented and continue to explore the use of a smartphone in combination with an AR-HMD. While their use for text selection is still rare, it has been investigated more broadly for other tasks.

### 2.2 Combining Handheld Devices and HMDs/Large Wall Displays

By combining handheld devices and HMDs, researchers try to make the most of the benefits of both [60]. On one hand, the handheld device brings a 2D high-resolution display that provides a multi-touch, tangible, and familiar interactive surface. On the other hand, HMDs provide a spatialized, 3D, and almost infinite work-space. With MultiFi [23], Grubert et al. showed that such a combination is more efficient than a single device for pointing and searching tasks. For a similar setup, Zhu and Grossman proposed a set of techniques and demonstrated how it can be used to manipulate 3D objects [60]. Similarly, Ren et al. [47] demonstrated how it can be used to perform windows management. Finally, in VESAD [44], Normand et al. used AR to directly extend the smartphone display.

Regarding the type of input provided by the handheld device, it is possible to only focus on using touch interactions, as it is proposed in Input Forager [2] and Dual-MR [35]. Waldow et al. compared the use of touch to perform 3D object manipulation to gaze and mid-air gestures and showed that touch was more efficient [53]. It is also possible to track the handheld device in space and allow for 3D spatial interactions. It has been done in DualCAD in which Millette and McGuffin used a smartphone tracked in space to create and manipulate shapes using both spatial interactions and touch gestures [41]. With ARPointer [49], Ro et al. proposed a similar system and showed it led to better performance for object manipulation than a keyboard and mouse combination as well as a combination of gaze

and mid-air gestures. When comparing the use of touch and spatial interaction with a smartphone, Budhiraja et al. showed that touch was preferred by participants for a pointing task [8], but Büschel et al. showed that spatial interaction was more efficient and preferred for a navigation task in 3D [9]. In both cases, Chen et al. showed that the interaction should be viewport-based and not world-based [17].

Overall, previous research showed a handheld device provides a good alternative input for augmented reality display in various tasks. In this paper, we focus on a text selection task, which has not been studied yet. It is not clear yet if only tactile interactions should be used on the handheld device or if it should also be tracked to provide spatial interactions. Thus, we propose the two alternatives in our techniques and compare them.

The use of handheld devices as input was also investigated in combination with large wall-displays. It is a use case close to the one presented in this paper as text is displayed inside a 2D virtual window. Campbell et al. studied the use of a Wiimote as a distant pointing device [10]. With a pointing task, the authors compared its use with an absolute mapping (i.e. raytracing) to a relative mapping, and showed that participants were faster with the absolute mapping. Vogel and Balakrishnan found similar results between the two mappings (with the difference that they directly tracked the hand), but only with large targets and when clutching was necessary [52]. They also found that participants had a lower accuracy with an absolute mapping. This lower accuracy for an absolute mapping with spatial interaction is also shown when compared with distant touch interaction of the handheld device as a trackpad, with the same task [5]. Jain et al. also compared touch interaction with spatial interaction, but with a relative mapping, and found that the spatial interaction was faster but less accurate [30]. The accuracy result is confirmed by a recent study from Siddhpuria et al. in which the authors also compared the use of absolute and relative mapping with the touch interaction, and found that the relative mapping is faster [50]. These studies were all done for a pointing task, and overall showed that using the handheld device as a trackpad (so with a relative mapping) is more efficient (to avoid clutching, one can change the transfer function [43]). In their paper, Siddhpuria et al. highlighted the fact that more studies needed to be done with a more complex task to validate their results. To our knowledge, this has been done only by Baldauf et al. with a drawing task, and they showed that spatial interaction with an absolute mapping was faster than using the handheld device as a trackpad without any impacts on the accuracy [5]. In this paper, we take a step in this direction and use a text selection task. Considering the result from Baldauf et al. we cannot assume that touch interaction will perform better.

### 2.3 Text Selection on Handheld Devices

Text selection has not been yet investigated with the combination of a handheld device and an AR-HMD, but it has been studied on handheld devices independently. Using a touchscreen, adjustment handles are the primary form of text selection techniques. However, due to the fat-finger problem [6], it can be difficult to modify the selection by one character. A first solution is to allow users to only select the start and the end of the selection as it is done in TextPin in which it is shown to be more efficient than the default technique [27]. Fuccella et al. [20] and Zhang et al. [59] proposed to use the keyboard area to allow the user to control the selection using gestures and showed it was also more efficient than the default technique. Ando et al. adapted the principle of shortcuts and associated different actions with the keys of the virtual keyboard that was activated with a modifier action performed after. In the first paper, the modifier was the tilting of the device [3], and in a second one, it was a sliding gesture starting on the key [4]. The latter was more efficient than the first one and the default technique. With BezelCopy [16], a gesture on the bezel of the phone allow for a first rough selection that can be refined after. Finally, other solutions used a non-traditional

smartphone. Le et al. used a fully touch-sensitive device to allow users to perform gestures on the back of the device [33]. Gaze N'Touch [48] used gaze to define the start and end of the selection. Goguey et al. explored the use of a force-sensitive screen to control the selection [22], and Eady and Girouard used a deformable screen to explore the use of the bending of the screen [19].

In this work, we choose to focus on commercially available smartphones, and we will not explore in this paper, the use of deformable, or fully touch-sensitive ones. Compared to the use of shortcuts, the use of gestures seems to lead to good performance and can be performed without looking at the screen (i.e. eye-free), which avoids transition between the AR virtual display and the handheld devices.

## 3 DESIGNING SMARTPHONE-BASED TEXT SELECTION IN AR-HMD

### 3.1 Proposed Techniques

Previous work used a smartphone as an input device to interact with virtual content in AR-HMD mainly in two ways — touch input from the smartphone and tracked the smartphone spatially like AR/VR controller. Similar work on wall-displays suggested that using the smartphone as a trackpad would be the most efficient technique, but this was tested with a pointing task (see related work). With a drawing task (which could be closer to a text selection task than a pointing task), spatial interaction was actually better [5].

Inspired by this, we propose four eyes-free text selection techniques for AR-HMD — two are completely based on mobile touchscreen interaction, whereas the smartphone needs to be tracked in mid-air for the latter two approaches to use spatial interactions. For spatial interaction, we choose a technique with an absolute mapping (*Raycasting*) and one with a relative one (*Spatial Movement*). The comparison between both in our case is not straightforward, previous results suggest that a relative mapping would have better accuracy, but an absolute one would be faster. For touch interaction, we choose to not have an absolute mapping, its use with a large virtual window could lead to a poor accuracy [43], and just have a technique that uses a relative mapping. In addition to the traditional use of the smartphone as a trackpad (*Continuous Touch*), we propose a technique that allows for a discrete selection of text (*Discrete Touch*). Such discrete selection mechanism has shown good results in a similar context for shape selection [30]. Overall, while we took inspiration from previous work for these techniques, they have never been assessed together for a text selection task.

To select text successfully using any of our proposed techniques, a user needs to follow the same sequence of steps each time. First, she moves the cursor, located on the text window in an AR display, to the beginning of the text to be selected (i.e., the first character). Then, she performs a double tap on the phone to confirm the selection of that first character. She can see on the headset screen that the first character got highlighted in yellow color. At the same time, she enters into the text selection mode. Next, she continues moving the cursor to the end position of the text using one of the techniques presented below. While the cursor is moving, the text is also getting highlighted simultaneously up to the current position of the cursor. Finally, she ends the text selection with a second double-tap.

#### 3.1.1 Continuous Touch

In continuous touch, the smartphone touchscreen acts as a trackpad (see Figure. 2(a)). It is an indirect pointing technique where the user moves her thumb on the touchscreen to change the cursor position on the AR display. For the mapping between display and touchscreen, we used a relative mode with clutching. As clutching may degrades performance [13], a control-display (CD) gain was applied to minimize it (see Section 3.2).

#### 3.1.2 Discrete Touch

This technique is inspired by the text selection with keyboard shortcuts available in both Mac [28] and Windows [25] OS. In this work, we tried to emulate a few keyboard shortcuts. We particularly considered imitating keyboard shortcuts related to the character, word, and line-level text selection. For example, in Mac OS, hold down ⇧ and pressing → or ← extends text selection one character to the right or left. Whereas hold down ⇧ + ⌥ and pressing → or ← allows users to select text one word to the right or left. To perform text selection to the nearest character at the same horizontal location on the line above or below, a user needs to hold down ⇧ and press ↑ or ↓ respectively. In discrete touch interaction, we replicated all these shortcuts using directional swipe gestures (see Figure. 2(b)). Left or right swipe can select text at both levels - word as well as character. By default, it works at the word level. The user performs a long-tap which acts as a toggle button to switch between word and character level selection. On the other hand, up or down swipe selects text at one line above or one line below from the current position. The user can only select one character/word/line at a time with its respective swipe gesture.

Note that, to select text using discrete touch, a user first positions the cursor on top of the starting word (not the starting character) of the text to be selected by touch dragging on the smartphone as described in the continuous touch technique. From a pilot study, we observed that moving the cursor every time to the starting word using discrete touch makes the overall interaction slow. Then, she selects that first word with the double-tap and uses discrete touch to select text up to the end position as described before.

#### 3.1.3 Spatial Movement

This technique emulates the smartphone as an air-mouse [1, 39] for AR-HMD. To control the cursor position on the headset screen, the user holds the phone in front of her torso, places her thumb on the touchscreen, and then she moves the phone in the air with small forearm motions in a plane that is perpendicular to the gaze direction (see Figure. 2(c)). While moving the phone, its tracked positional data in XY coordinates get translated into the cursor movement in XY coordinates inside a 2D window. When a user wants to stop the cursor movement, she simply lifts her thumb from the touchscreen. Thumb touch-down and touch-release events define the start and stop of the cursor movement on the AR display. The user determines the speed of the cursor by simply moving the phone faster and slower accordingly. We applied CD-gain between the phone movement and the cursor displacement on the text window (see Section 3.2).

#### 3.1.4 Raycasting

Raycasting is a popular interaction technique in AR/VR environments to select 3D virtual objects [7, 42]. In this work, we developed a smartphone-based raycasting technique for selecting text displayed on a 2D window in AR-HMD (see Figure. 2(d)). A 6 DoF tracked smartphone was used to define the origin and orientation of the ray. In the headset display, the user can see the ray in a straight line appearing from the top of the phone. By default, the ray is always visible to users in AR-HMD as long as the phone is being tracked properly. In raycasting, the user needs to do small angular wrist movements for pointing on the text content using the ray. Where the ray hits on the text window, the user sees the cursor there. Compared to other proposed methods, raycasting does not require clutching as it allows direct pointing to the target. The user confirms the target selection on the AR display by providing a touch input (i.e., double-tap) from the phone.

### 3.2 Implementation

To prototype our proposed interaction techniques, we used a Microsoft HoloLens 2 ($42° \times 29°$ screen) as an AR-HMD device and a

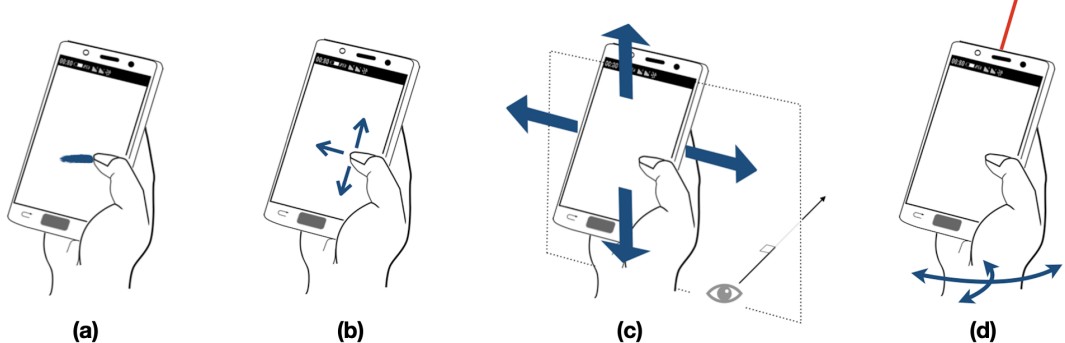

Figure 2: Illustrations of our proposed interaction techniques: (a) continuous touch; (b) discrete touch; (c) spatial movement; (d) raycasting.

| Techniques | $CD_{Max}$ | $CD_{Min}$ | $\lambda$ | $V_{inf}$ |
|---|---|---|---|---|
| Continuous Touch | 28.34 | 0.0143 | 36.71 | 0.039 |
| Spatial Movement | 23.71 | 0.0221 | 32.83 | 0.051 |

Table 1: Logistic function parameter values for continuous touch and spatial movement interaction. The unit of $CD_{Max}$ and $CD_{Min}$ is in mm/mm, whereas $\lambda$ is in sec/mm and $V_{inf}$ is in mm/sec.

OnePlus 5 as a smartphone. For spatial movement and raycasting interactions, real-time pose information of the smartphone is needed. An OptiTrack[3] system with three Flex-13 cameras was used for accurate tracking with low latency. To bring the hololens and the smartphone into a common coordinate system, we attached passive reflective markers to them and did a calibration between hololens space and optitrack space.

In our software framework, the AR application running on HoloLens was implemented using Unity3D (2018.4) and Mixed Reality Toolkit[4]. To render text in HoloLens, we used TextMesh-Pro. A Windows 10 workstation was used to stream tracking data to HoloLens. All pointing techniques with the phone were also developed using Unity3D. We used UNet[5] library for client-server communications between devices over the WiFi network.

For continuous touch and spatial movement interactions, we used a generalized logistic function [43] to define the CD-gain between the move events either on the touchscreen or in the air and the cursor displacement in the AR display:

$$CD(v) = \frac{CD_{Max} - CD_{Min}}{1 + e^{-\lambda \times (v - V_{inf})}} + CD_{Min} \qquad (1)$$

$CD_{Max}$ and $CD_{Min}$ are the asymptotic maximum and minimum amplitudes of CD gain and $\lambda$ is a parameter proportional to the slope of the function at $v = V_{inf}$ with $V_{inf}$ a inflection value of the function. We derived initial values from the parameters of the definitions from Nancel et al. [43], and then empirically optimized for each technique. The parameters were not changed during the study for individual participants. The values are summarized in Table 1.

In discrete touch interaction, we implemented up, down, left, and right swipes by obtaining touch position data from the phone. We considered a 700 msec time window for detecting a long-tap event after doing a pilot test with four users from our lab. Users get vibration feedback from the phone when they perform long-tap successfully. They also receive vibration haptics while double-tapping to start and end the text selection in all interaction techniques.

Note that, there is no haptic feedback for swipes. With each swipe movement, they can see that texts are getting highlighted in yellow color. This acts as visual feedback by default for touch swipes.

In the spatial movement technique, we noticed that the phone moves slightly during the double-tap event. This results in a slight unintentional cursor movement. To reduce that, we suspended cursor movement for 300 msec when there is any touch event on the phone screen. We found this value after doing trial and error with different values ranging from 150 msec to 600 msec.

In raycasting, we applied the 1€ Filter [12] with $\beta = 80$ and min-cutoff = 0.6 at the ray source to minimize jitter and latency which usually occur due to both hand tremor and double-tapping [55]. We tuned these two parameters by following the instruction mentioned in the 1€ Filter implementation website[6]. We set the ray length to 8 meters by default. The user sees the full length of the ray when it is not hitting the text panel.

## 4 EXPERIMENT

To assess the impact of the different characteristics of these four interaction techniques we perform a comparative study with a text selection task while users are standing up. Particularly, we are interested to evaluate the performance of these techniques in terms of task completion time, accuracy, and perceived workload.

### 4.1 Participants and Apparatus

In our experiment, we recruited 20 unpaid participants (P1-P20) (13 males + 7 females) from a local university campus. Their ages ranged from 23 to 46 years (mean = 27.84, SD = 6.16). Four were left-handed. All were daily users of smartphones and desktops. With respect to their experience with AR/VR technology, 7 participants ranked themselves as an expert because they are studying and working on the same field, 4 participants were beginners as they played some games in VR, while others had no prior experience. They all had either normal or corrected-to-normal vision. We used the apparatus and prototype described in Subsection 3.2.

### 4.2 Task

In this study, we ask participants to perform a series of text selections using our proposed techniques. Participants were standing up for the entire duration of the experiment. We reproduce different realistic usage by varying the type of text selection to do, like the selection of a word, a sentence, a paragraph, etc. Figure 3 shows all the types of text selection that were asked to the participants. Concretely, the experiment scene in HoloLens consisted of two vertical windows of 102.4 cm $\times$ 57.6 cm positioned at a distance of 180 cm from the headset at the start of the application (i.e., its visual size was 31.75° $\times$ 18.1806°). The windows were anchored in the

---

[3]https://optitrack.com/
[4]https://github.com/microsoft/MixedRealityToolkit-Unity
[5]https://docs.unity3d.com/Manual/UNet.html

[6]https://cristal.univ-lille.fr/ casiez/1euro/

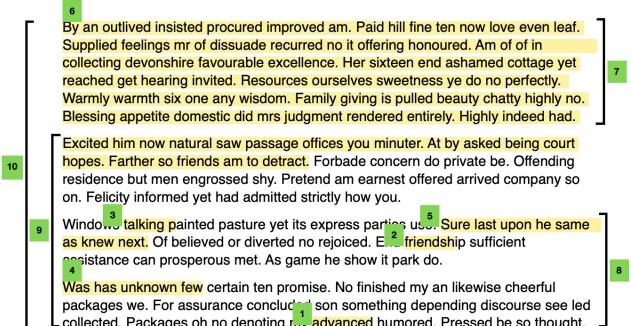

Figure 3: Text selection tasks used the experiments: (1) word (2) sub-word (3) word to a character (4) four words (5) one sentence (6) paragraph to three sentences (7) one paragraph (8) two paragraphs (9) three paragraphs (10) whole text.

world coordinate. These two panels contain the same text. Participants are asked to select the text in the action panel (left panel in Figure 1(b)) that is highlighted in the instruction panel (right panel in Figure 1(b)). The user controls a cursor (i.e., a small circular dot in red color as shown in Figure 1(b)) using one of the techniques on the smartphone. Its position is always bounded by the window size. The text content was generated by Random Text Generator[7] and was displayed using the *Liberation Sans* font with a font-size of 25 pt (to allow a comfortable viewing from a few meters).

### 4.3 Study Design

We used a within-subject design with 2 factor: 4 INTERACTION TECHNIQUE (*Continuous Touch*, *Discrete Touch*, *Spatial Movement*, and *Raycasting*) × 10 TEXT SELECTION TYPE (shown in Figure 3) × 20 participants = 800 trials. The order of INTERACTION TECHNIQUE was counterbalanced across participants using a Latin Square. The order of TEXT SELECTION TYPE is randomized in each block for each INTERACTION TECHNIQUE (but same for each participant).

### 4.4 Procedure

We welcomed participants upon arrival. They were asked to read and sign the consent form, fill out a pre-study questionnaire to collect demographic information and prior AR/VR experience. Next, we gave them a brief introduction to the experiment background, hardware, the four interaction techniques, and the task involved in the study. After that, we helped participants to wear HoloLens comfortably and complete the calibration process for their personal interpupillary distance (IPD). For each block of INTERACTION TECHNIQUE, participants completed a practice phase followed by a test session. During the practice, the experimenter explained how the current technique worked, and participants were encouraged to ask questions. Then, they had time to train themselves with the technique until they were fully satisfied, which took around 7 minutes on average. Once they felt confident with the technique, the experimenter launched the application for the test session. They were instructed to do the task as quickly and accurately as possible in a standing condition. To avoid noise due to participants using either one or two hands, we asked to only use their dominant hand.

At the beginning of each trial in the test session, the text to select was highlighted in the instruction panel. Once they were satisfied with their selection, participants had to press a dedicated button on the phone screen to get to the new task. They were allowed to use their non-dominant hand only to press this button. At the end of each block of INTERACTION TECHNIQUE, they answered a NASA-TLX questionnaire [24] on iPad, and moved to the next condition.

---

[7]http://randomtextgenerator.com/

At the end of the experiment, we asked participants a questionnaire in which they had to rank techniques by speed, accuracy, and overall preference and performed an informal post-test interview.

The entire experiment took approximately 80 minutes in total. Participants were allowed to take breaks between sessions during which they could sit and encourage to comment at any time during the experiment. To respect COVID-19 safety protocol, participants wore FFP2 mask and maintained a 1-meter distance with the experimenter at all times.

### 4.5 Measures

We recorded completion time as the time taken to select the text from its first character to the last character, which is the time difference between the first and second double-tap. If they selected more or less characters than expected, the trial was considered wrong. We then calculated the error rate as the percentage of wrong trials for each condition. Finally, as stated above, participants filled a NASA TLX questionnaire to measure the subjective workload of each INTERACTION TECHNIQUE, and their preference was measured using a ranking questionnaire at the end of the experiment.

### 4.6 Hypotheses

In our experiment, we hypothesized that:

**H1.** *Continuous Touch*, *Spatial Movement*, and *Raycasting* will be faster than *Discrete Touch* because a user needs to spend more time for multiple swipes and do frequent mode switching to select text at the character/word/sentence level.

**H2.** *Discrete Touch* will be more mentally demanding compared to all other techniques because the user needs to remember the mapping between swipe gestures and text granularity, as well as the long-tap for mode switching.

**H3.** The user will perceive that *Spatial Movement* will be more physically demanding as it involves more forearm movements.

**H4.** The user will make more errors in *Raycasting*, and it will be more frustrating because double-tapping for target confirmation while holding the phone in one hand will introduce more jitter [55].

**H5.** Overall, *Continuous Touch* would be the most preferred text selection technique as it works similarly to the trackpad which is already familiar to users.

## 5 RESULT

To test our hypothesis, we conducted a series of analyses using IBM SPSS software. Shapiro-Wilk tests showed that the task completion time, total error, and questionnaire data were not normally distributed. Therefore, we used the Friedman test with the interaction technique as an independent variable to analyze our experimental data. When significant effects were found, we reported post hoc tests using the Wilcoxon signed-rank test and applied Bonferroni corrections for all pair-wise comparisons. We set an $\alpha = 0.05$ in all significance tests. Due to a logging issue, we had to discard one participant and did the analysis with 19 instead of 20 participants.

### 5.1 Task Completion Time

There was a statistically significant difference in task completion time depending on which interaction technique was used for text selection [$\chi^2$ (3) = 33.37, $p < .001$] (see Figure 4(a)). Post hoc tests showed that *Continuous Touch* [M = 5.16, SD = 0.84], *Spatial Movement* [M = 5.73, SD = 1.38], and *Raycasting* [M = 5.43, SD = 1.66] were faster than *Discrete Touch* [M = 8.78, SD = 2.09].

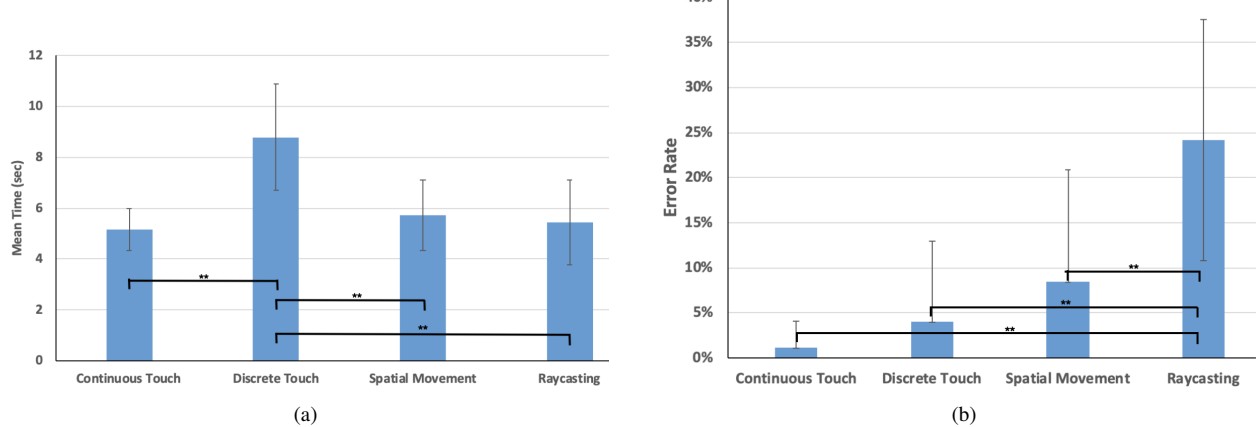

(a)

(b)

Figure 4: (a) Mean task completion time for our proposed four interaction techniques. Lower scores are better. (b) Mean error rate of interaction techniques. Lower scores are better. Error bars show 95% confidence interval. Statistical significances are marked with stars (**: $p < .01$ and *: $p < .05$).

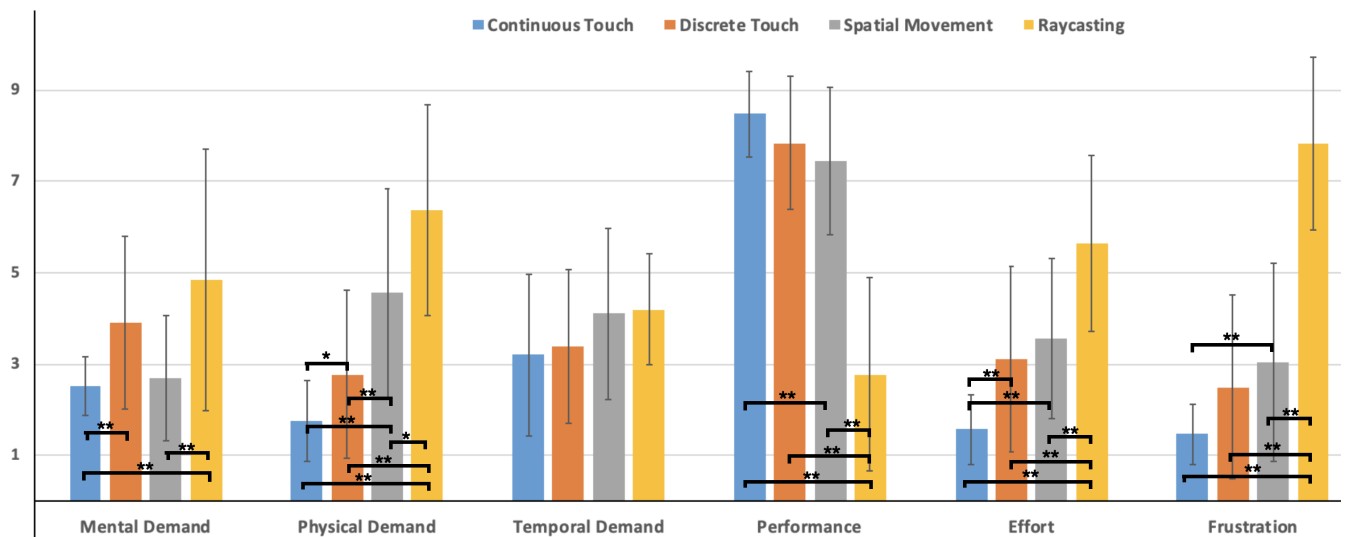

Figure 5: Mean scores for the NASA-TLX task load questionnaire which are in range of 1 to 10. Lower marks are better, except for performance. Error bars show 95% confidence interval. Statistical significances are marked with stars (**: $p < .01$ and *: $p < .05$).

## 5.2 Error Rate

We found significant effects of the interaction technique on error rate [$\chi^2$ (3) = 39.45, $p < .001$] (see Figure 4(b)). Post hoc tests showed that *Raycasting* [M = 24.21, SD = 13.46] was more error-prone than *Continuous Touch* [M = 1.05, SD = 3.15], *Discrete Touch* [M = 4.73, SD = 9.05], and *Spatial Movement* [M = 8.42, SD = 12.58].

## 5.3 Questionnaires

For NASA-TLX, we found significant differences for mental demand [$\chi^2$ (3) = 9.65, $p = .022$], physical demand [$\chi^2$ (3) = 29.75, $p < .001$], performance [$\chi^2$ (3) = 40.14, $p < .001$], frustration [$\chi^2$ (3) = 39.53, $p < .001$], and effort [$\chi^2$ (3) = 32.69, $p < .001$]. Post hoc tests showed that *Raycasting* and *Discrete Touch* had significantly higher mental demand compared to *Continuous Touch* and *Spatial Movement*. On the other hand, physical demand was lowest for *Continuous Touch*, whereas users rated significantly higher physical demand for *Raycasting* and *Spatial Movement*. In terms of performance, *Raycasting* was rated significantly lower than the

other techniques. *Raycasting* was also rated significantly more frustrating. Moreover, *Continuous Touch* was least frustrating and better in performance than *Spatial Movement*. Figure 5 shows a bar chart of the NASA-TLX workload sub-scales for our experiment.

For ranking questionnaires, there were significant differences for speed [$\chi^2$ (3) = 26.40, $p < .001$], accuracy [$\chi^2$ (3) = 45.5, $p < .001$], and preference [$\chi^2$ (3) = 38.56, $p < .001$]. Post hoc test showed that users ranked *Discrete Touch* as the slowest and *Raycasting* as the least accurate technique. The most preferred technique was *Continuous Touch* whereas *Raycasting* was the least. Users also favored *Discrete Touch* as well as *Spatial Movement* based text selection approach. Figure 6 summarises participants responses for ranking questionnaires.

## 6 DISCUSSION & DESIGN IMPLICATIONS

Our results suggest that *Continuous Touch* is the technique that was preferred by the participants (confirming **H5**). It was the least physically demanding technique and the less frustrating one. It was also

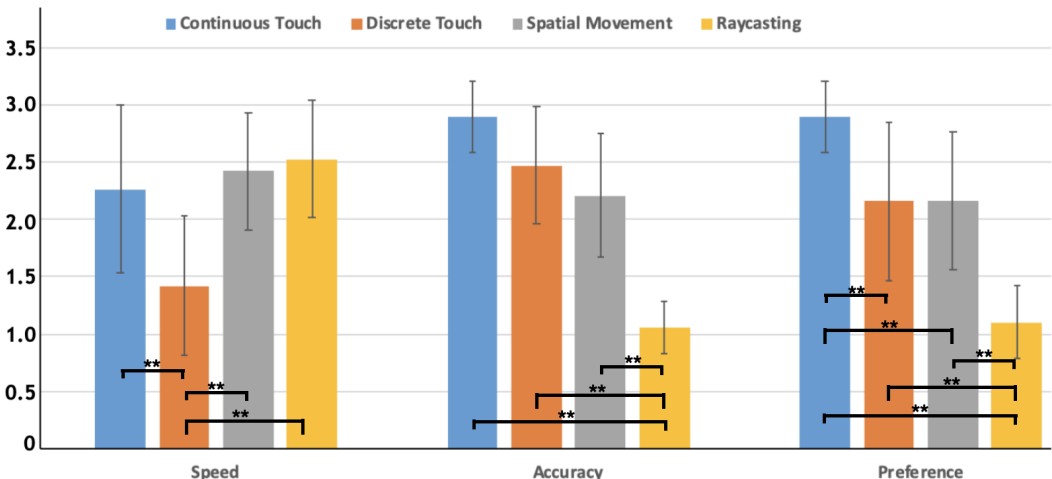

Figure 6: Mean scores for the ranking questionnaire which are in 3 point likert scale. Higher marks are better. Error bars show 95% confidence interval. Statistical significances are marked with stars (**: $p < .01$ and *: $p < .05$).

more satisfying regarding performance than the two spatial ones (*Raycasting* and *Spatial Movement*). Finally, it was less mentally demanding than *Discrete Touch* and *Raycasting*. Participants pointed out that this technique was simple, intuitive, and familiar to them as they are using trackpad and touchscreen every day. During the training session, we noticed that they took the least time to understand its working principle. In the interview, P8 commented, *"I can select text fast and accurately. Although I noticed a bit of overshooting in the cursor positioning, it can be adjusted by tuning CD gain"*. P17 said, *"I can keep my hands down while giving input to select text in AR display. This gives me more comfort"*.

On the other hand, *Raycasting* was the least preferred technique and led to the lowest task accuracy (confirming **H4**). Participants (irrespective of experienced and inexperienced) were also the least satisfied with their performance using this technique. This can be explained by the fact that it was the most physically demanding and the most frustrating. Finally, it was more mentally demanding than *Continuous Touch* and *Spatial Movement*. In their comments, participants reported about the lack of stability due to the one-handed phone holding posture. Some participants complained that they felt uncomfortable to hold this OnePlus 5 phone in one hand as it was a bit bigger compared to their hand size. This introduced even more jitter for them in *Raycasting* while double-tapping for target confirmation. P10 commented, *"I am sure I will perform Raycasting with fewer errors if I can use my both hands to hold the phone"*. Moreover, from the logged data, we noticed that they made more mistakes when the target character was positioned inside a word rather than either at the beginning or at the end, which was confirmed in the discussion with participants.

As we expected, *Discrete Touch* was the slowest technique (confirming **H1**), but was not the most mentally demanding, as it was only more demanding than *Continuous Touch* (rejecting **H2**). It is also more physically demanding than *Continuous Touch*, but less than *Spatial Movement* and *Raycasting*. Several participants mentioned that it is excellent for the short word to word or sentence to sentence selection, but not for long text as multiple swipes are required. They also pointed out that performing mode switching with a long-tap of 700 msec was a bit tricky and lost some time there during text selection. Although they got better with it over time, still they are uncertain to do it successfully in one attempt. To improve this mode switching, one participant suggested using a triple tap for mode switching instead of a long-tap.

Finally, contrary to our expectation, *Spatial Movement* was not

the most physically demanding technique, as it was less demanding than *Raycasting* but more than *Continuous Touch* and *Discrete Touch* (rejecting **H3**). It was also less mentally demanding than *Raycasting* and led to less frustration. However, it led to more frustration and participants were less satisfied with their performance with this technique than with *Continuous Touch*. According to participants, with this technique, moving the forearm needs physical effort undoubtedly, but they only need to move it for a very short distance which was fine for them. From the user interview, we came to know that they did not use much clutching (less than with *Continuous Touch*). P13 mentioned, *"In Spatial Movement, I completed most of the tasks without using clutching at all"*.

Overall, our results suggest that between touch and spatial interactions, it would be better to use touch for text selection, which confirms findings from Siddhpuria et al. for pointing tasks [50]. *Continuous Touch* was overall preferred, faster, and less demanding than *Discrete Touch*, which goes against results from the work by Jain et al. for shape selection [30]. Such difference can be explained by the fact that with text selection, there is a minimum of two levels of discretization (characters and words), which makes it mentally demanding. It can also be explained by the high number of words (and even more characters) in a text, contrary to the number of shapes in Jain et al. experiment. This led to a high number of discrete actions for the selection, and thus, a higher physical demand. However, surprisingly, most of the participants appreciated the idea of *Discrete Touch*. If a tactile interface is not available on the handheld device, our results suggest to use a spatial interaction technique that uses a relative mapping, as we did with *Spatial Movement*. We could not find any differences in time, contrary to the work by Campbell et al. [10], but it leads to fewer errors, which confirms what was found by Vogel and Balakrishnan [52]. It is also less physically and mentally demanding and leads to less frustration than an absolute mapping. On the technical side, a spatial interaction technique with a relative mapping can be easily achieved without an external sensor (as it was done for example by Siddhpuria et al. [50]).

## 7 LIMITATIONS

There were two major limitations. First, we used an external tracking system which limits us to lab study only. As a result, it is difficult to understand the social acceptability of each technique until we consider the real-world on-the-go situation. However, technical

progress in inside-out tracking[8] means that it will be possible, soon, to have smartphones that can track themselves accurately in 3D space. Second, some of our participants had difficulties holding the phone in one hand because the phone was a bit bigger for their hands. They mentioned that although they were trying to move their thumb faster in continuous touch and discrete touch interactions, they were not able to do it comfortably due to the afraid of dropping the phone. This bigger phone size also influenced their raycasting performance particularly when they need to do a double-tap for target confirmation. Hence, using one phone size for all was an important constraint in this experiment.

## 8 CONCLUSION AND FUTURE WORK

In this research, we investigated the use of a smartphone as an eyes-free interactive controller to select text in augmented reality head-mounted display. We proposed four interaction techniques: 2 that use the tactile surface of the smartphone (continuous touch and discrete touch), and two that track the device in space (spatial movement and raycasting). We evaluated these four techniques in a text selection task study. The results suggested that techniques using the tactile surface of the device are more suited for text selection than spatial one, continuous touch being the most efficient. If a tactile surface was not available, it would be better to use a spatial technique (i.e. with the device tracked in space) that uses a relative mapping between the user gesture and the virtual screen, compared to a classic raycasting technique that uses an absolute mapping.

In this work, we have focused on interaction techniques based on smartphone inputs. This allowed us to better understand which approach should be favored in that context. In the future, it would be interesting to explore a more global usage scenario such as a text editing interface in AR-HMD using smartphone-based input where users need to perform other interaction tasks such as text input and commands execution simultaneously. Another direction to future work is to compare phone-based techniques to other input techniques like hand tracking, head/eye gaze, and voice commands. Furthermore, we only considered standing condition, but it would be interesting to study text selection performance while walking.

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
