# OpenReview forum: "Exploring Smartphone-enabled Text Selection in AR-HMD"
_graphicsinterface.org/Graphics_Interface/2021/Conference/Second_Cycle — GI 2021_

### Official Review · Reviewer_fJxi · 2021-05-03
**Review of "Exploring Smartphone-enabled Text Selection in AR-HMD"**

**Rating:** 6
**Confidence:** 5

**Review:**

This paper studies the use of a smartphone to support text selection in AR-HMD. To this end, the authors make an empirical evaluation of four distinct selection mechanisms: continuous touch, discrete touch, spatial movement, and raycasting. While none of these selection methods are novel, the contribution of the paper lies in making a fair and objective comparison with respect to usability/usefulness in the AR-HMD setting. Based on the results of the study, continuous touch was accurate, fast, and overall rated well.

The paper is well-written, the user study is appropriately designed and analyzed. However, there are some clarifications that need to be addressed that are detailed below.

In the implementation section, various design decisions are not explained and justified: e.g., the delay time for the cursor movement (300 ms) in spatial movement technique, or filter parameters in raycasting, and long press (700 ms) in discrete touch. Although authors note in passing that these values were selected empirically, a more detailed explanation of the selection procedure is needed.

The vast majority of the participants are experts in AR/VR who study or work in the same area. It would be interesting to see the performance comparison (and detailed analysis) of experienced and inexperienced participants. I assume experienced participants would outperform inexperienced in some common techniques in AR/VR (e.g., raycasting).

Based on the results, raycasting was the least preferred and the most frustrating technique. The authors used OnePlus 5 as a ray source, and participants had to perform angular wrist movements while holding the phone for text selection. Moving wrist constantly with such a big and heavy phone may cause fatigue and might result in a poor experience for the participants. This claim is supported by P10, who wanted to hold the phone with both hands to perform better. Although, it was stated as one of the limitations (sec 7), I believe it should be more prominently discussed as raycasting is one of the leading selection mechanisms in VR. While the conclusion of the paper might suggest that raycasting is “frustrating to use”, it should be understood as “raycasting using a heavy phone is frustrating to use”. Making this distinction would be helpful.

The discrete touch was the slowest technique, I assume it happened due to the following factors:
* It was the only technique that required switching between word and character level
* It forced delay time (700 msec), to switch between levels

Based on this, one might consider that forced delay time has influenced the overall speed of the technique. Please address it.

In the discussion section, authors mostly analyze user preference. However, it would be interesting to see how each method performed with different task types. E.g., whether discrete touch outperformed others in 1st, 2nd, and 3rd text selection tasks, whether raycasting was better for 7th, 8th, 9th, and 10th tasks (some graphs?). Such additional details would be helpful for future researchers to make the design decisions.

In summary, I find this work interesting and promising. However, some details of design decisions are missing, if the authors elaborate on that, it will be a solid contribution to the conference.

Latex, style, and writing issues.

The dashes throughout the text are typesetted improperly. It should be "---", but authors always use "-". Some examples:
On page 1: "input - to touch or spatial" must be "input---to touch or spatial". Similar mistakes are throughout the paper, please fix them.

Similarly, authors should consider instructing Latex on non-sentence-ending periods. For example, "from Siddphuria et al. for" should be typeset as "from Siddphuria et al.\ for". This would eliminate unnecessary whitespaces.

Use lower case inside of the parenthesis. For example "(See Related Work)" of section 3.1 must be "(see related work)".

Section 2.1:
"Very few research focused" => "Limited research has been focused"

Section 3.1.3:
CD-gain is used before its formal definition. Please correct.

In Abstract:
I believe instead of using dash after "all using a smartphone", a semicolon is more appropriate. However, if authors decide to leave dash, please typeset it properly (i.e. as "---").

---

### Official Review · Reviewer_XQpG · 2021-05-04
**Exploring Smartphone-enabled Text Selection in AR-HMD**

**Rating:** 6
**Confidence:** 3

**Review:**

This paper explores 4 ways of using a smart phone to select text elements displayed through an HMD in an AR/VR environment. I did not find much innovation in the user interaction techniques proposed in these 4 ways.  The main work is the formal user study for evaluating the 4 proposed interaction techniques.  The user study seems to have been conducted well and systematically, with 25 users, each interacting for over 4 hours with the text selection techniques Given the current pandemic related situation we are in, conducting such a user study is commendable.  It does not appear to be a unique study, though. In that sense from a scientific perspective the contribution does not seem significant. My one concern is that the authors are considering the text selection task in isolation. While the authors have identified problems with other selection techniques, such as use of voice commands not being usable in all situations,  they have not indicated any solutions for the tasks beyond text selection. After text selection, if we have to use one of the other solutions for tasks, like text editing, then what have we gained? In my opinion, it would be necessary to  see different tasks being performed on the selected text to judge the true value of the proposed techniques.

On the writing front, there a few typos and language/expression problems, gender mix-up, like using a female reference for the user (fine with that), but then saying his, etc. The text needs some careful reading, but overall  these are minor problems which do not detract the reader from understanding the material.

---

### Official Review · Reviewer_ZHgx · 2021-05-05
**Good study, but treads very familiar ground**

**Rating:** 5
**Confidence:** 4

**Review:**

This paper proposes four text-selection techniques for head-mounted augmented reality, which use a paired smartphone as either a pointer or a touchpad. A user study on 20 participants suggests that using the smartphone as a touchpad, using continuous touch input, performs the best.

There is a great deal of research literature on selection with pointers and handheld devices for remote displays, which are largely similar to AR-HMDs in this use-case. The authors do a good job of summarizing related work, but it is certainly a crowded space. Although there is perhaps less work on text selection specifically, text selection would seem to be a fairly niche use-case considering that text editing is not a primary task carried out in these types of environments. In this sense, the scope of this paper's results would seem to be rather limited. In particular, compared to the existing research literature, most notably the study from Nancel et al. [42], this paper does not seem to add much.

The study is reasonably well-executed, although the raycasting implementation seemed to suffer from very poor accuracy (nearly 25% error rate). This can largely be explained by the need to hit *very* small targets (i.e. letter boundaries) from far away, a task that direct raycasting is traditionally poor at (without extra assistance, e.g. snapping to word boundaries or C-D gain adjustments). The results are well-explained and analyzed.

Overall, I am a bit on the fence about this paper. The study and implementation seem to be reasonably well done, and the best-performing technique seems useful. However, it is also not a very novel technique, and the paper treads very familiar ground compared to existing works on pointing at remote displays. Hence, I do not feel that it has a very strong contribution to the literature.

---

### Meta-Review · Area_Chair_Zy32 · 2021-05-04

**Recommendation:** Accept
**Confidence:** 3

**Metareview:**

The main contribution in this paper is a user study to test certain hypotheses about 4 interaction methods to use  a smartphone for text selection in a AR/VR environment. The methods themselves have no significant innovation,  but the user study is well designed and conducted systematically. The results have also been statistically analyzed, and conclusions about acceptance/rejection of hypotheses made. Overall as a paper in user studies this is a good contribution. The authors are requested to take careful note  of the detailed comments from the reviewers and revise the final paper accordingly.

---

### Decision · Program_Chairs · 2021-05-08

Accept